# Nourin-Dependent miR-137 and miR-106b: Novel Early Inflammatory Diagnostic Biomarkers for Unstable Angina Patients

**DOI:** 10.3390/biom11030368

**Published:** 2021-02-28

**Authors:** Salwa A. Elgebaly, Robert H. Christenson, Hossam Kandil, Nashwa El-Khazragy, Laila Rashed, Beshoy Yacoub, Heba Eldeeb, Mahmoud Ali, Roshanak Sharafieh, Ulrike Klueh, Donald L. Kreutzer

**Affiliations:** 1Research & Development, Nour Heart, Inc., Vienna, VA 22180, USA; 2Department of Surgery, School of Medicine, UConn Health, Farmington, CT 06032, USA; rsharafieh@uchc.edu (R.S.); kreutzer@uchc.edu (D.L.K.); 3Department of Pathology, University of Maryland School of Medicine, Baltimore, MD 21201, USA; rchristenson@som.umaryland.edu; 4Department of Cardiology, Kasr Alainy Faculty of Medicine, Cairo University, Cairo 11562, Egypt; Hossamkandil@kasralainy.edu.eg (H.K.); beshoy.ayuob@gmail.com (B.Y.); hebamostafakamel@gmail.com (H.E.); mahmoudbe1@hotmail.com (M.A.); 5Department of Clinical Pathology-Hematology, Ain Shams Medical Research Institute (MASRI), Faculty of Medicine, Ain Shams University, Cairo 11566, Egyp; nashwaelkhazragy@med.asu.edu.eg; 6Department of Biochemistry and Molecular Biology, Kasr Alainy Faculty of Medicine, Cairo University, Cairo 11562, Egypt; lailarashed@kasralainy.edu.eg; 7Cell & Molecular Tissue Engineering, LLC Farmington, CT 06032, USA; klueh@wayne.edu; 8Integrative Biosciences Center (IBio), Department of Biomedical Engineering, Wayne State University, Detroit, MI 48202, USA

**Keywords:** unstable angina, Nourin, miRNAs, inflammatory diagnostic biomarkers, reversible myocardial ischemia, acute coronary syndromes

## Abstract

Background: Currently, no blood biomarkers exist that can diagnose unstable angina (UA) patients. Nourin is an early inflammatory mediator rapidly released within 5 min by reversible ischemic myocardium, and if ischemia persists, it is also released by necrosis. Nourin is elevated in acute coronary syndrome (ACS) patients but not in symptomatic noncardiac and healthy subjects. Recently, circulating microRNAs (miRNAs) have been established as markers of disease, including cardiac injury and inflammation. Objectives: To profile and validate the potential diagnostic value of Nourin-dependent *miR-137* (marker of cell damage) and *miR-106b-5p* (marker of inflammation) as early biomarkers in suspected UA patients and to investigate the association of their target and regulating genes. Methods: Using Nourin amino acid sequence, an integrated bioinformatics analysis was conducted. Analysis indicated that Nourin is a direct target for *miR-137* and *miR-106b-5p* in myocardial ischemic injury. Two linked molecular networks of lncRNA/miRNAs/mRNAs were also retrieved, including *CTB89H12.4/miR-137/FTHL-17* and *CTB89H12.4*/*miR-106b-5p*/*ANAPC11*. Gene expression profiling was assessed in serum samples collected at presentation to an emergency department (ED) from: (1) UA patients (*n* = 30) (confirmed by invasive coronary angiography with stenosis greater than 50% and troponin level below the clinical decision limit); (2) patients with acute ST elevation myocardial infarction (STEMI) (*n* = 16) (confirmed by persistent ST-segment changes and elevated troponin level); and (3) healthy subjects (*n* = 16). Results: Gene expression profiles showed that *miR-137* and *miR-106b-5p* were significantly upregulated by 1382-fold and 192-fold in UA compared to healthy, and by 2.5-fold and 4.6-fold in STEMI compared to UA, respectively. Healthy subjects showed minimal expression profile. Receiver operator characteristics (ROC) analysis revealed that the two miRNAs were sensitive and specific biomarkers for assessment of UA and STEMI patients. Additionally, Spearman’s correlation analysis revealed a significant association of miRNAs with the associated mRNA targets and the regulating lncRNA. Conclusions: Nourin-dependent gene expression of *miR-137* and *miR-106b-5p* are novel blood-based biomarkers that can diagnose UA and STEMI patients at presentation and stratify severity of myocardial ischemia, with higher expression in STEMI compared to UA. Early diagnosis of suspected UA patients using the novel Nourin biomarkers is key for initiating guideline-based therapy that improves patients’ health outcomes.

## 1. Introduction

Acute coronary syndromes encompass a spectrum of ischemic conditions that include acute myocardial infarction (AMI) with ST-segment elevation (STEMI) or without (NSTEMI) and unstable angina [1,2]. Currently, patients with chest pain are assessed by risk score algorithms [3,4], electrocardiogram (ECG), cardiac enzymes, and occasionally coronary computed tomographic angiography [5,6]. However, the diagnosis still has major challenges especially for patients with atypical symptoms and completely normal or dynamic come-and-go ECG changes [7]. Although the introduction of high-sensitive cardiac troponins has improved the detection rate of NSTEMI, in the absence of myocardial necrosis, patients with unstable angina (UA) often undergo a lengthy assessment in a hospital emergency department (ED) or require hospital admissions [8]. Additionally, to date, no inflammatory blood-based biomarkers exist that can quickly diagnose reversible ischemic injury in UA patients [9]. Thus, there is a clear need to identify and validate new blood-based biomarkers to identify reversible ischemic injury as seen in UA patients before progressing to necrosis.

Elgebaly et al. [10,11,12,13,14,15] reported that Nourin is a 3 KDa formyl peptide that is released within five minutes by human and animal hearts in response to ischemic injury. Nourin is unique because of its rapid release by reversible ischemic myocardium when cells are initially injured, but still alive and not dead. If ischemia persists, Nourin is also released by necrotic cells. Nourin is a potent inflammatory mediator, and its release is associated with post-ischemic cardiac inflammation in early ischemia/reperfusion animal models of AMI, cardiopulmonary bypass surgery, as well as heart failure. A schematic diagram illustrating the release of the leukocyte chemotactic factor Nourin by ischemic myocardium, followed by leukocyte recruitment and cardiac inflammation, is presented in Figure 1. In our earlier publications, Nourin was referred to as cardiac-derived leukocyte chemotactic factor. Nourin was purified from cardioplegic solutions collected during cardiac arrest (reversible myocardial ischemia) from cardiopulmonary bypass patients when they underwent coronary revascularization [11], and its amino acid sequence was determined. As a formyl peptide, Nourin binds to formyl peptide receptor (FPR) on leukocytes and vascular endothelial cells (VECs). A number of competitive antagonists were identified, including cyclosporin H, spinorphin, t-Boc-Phe-D.Leu-Phe-D.Leu-Phe, and soluble FPR fragment 17 aa loop peptide, which inhibited Nourin activity and tissue inflammation. Additionally, the bioenergetic compound, Cyclocreatine Phosphate (CCrP) prevented ischemic injury by preserving elevated levels of cellular adenosine triphosphate (ATP) during ischemia, thus reducing Nourin intracellular formation and circulating levels, resulting in reduction of early reperfusion cardiac inflammation and restoration of contractile function in animal models of AMI, global cardiac arrest, cardiopulmonary bypass, heart transplantation, as well as heart failure [10,13]. As a potent inflammatory mediator, Nourin stimulates leukocyte chemotaxis and VEC migration and activates human monocytes, neutrophils, and VECs to express high levels of cytokine storm mediators, digestive enzymes, and free radicals, including tumor necrosis factor-α (TNF-α, a key stimulant of apoptosis), interleukin 1β (IL-1β), interleukin 8 (IL-8), leukocyte-endothelial cell adhesion molecule 1 (LECAM-1), intercellular adhesion molecule 1 (ICAM-1), and endothelial-leukocyte adhesion molecule 1 (ELAM-1), as well as collagenase type IV, N-acetyl-B-glucosaminidase, gelatinases, and superoxide anion [10]. Several of these mediators were reported by other investigators to play a crucial role in recruiting and activating neutrophils during post-ischemic early reperfusion, and thus may contribute to the no-reflow phenomenon. Furthermore, since the above-described Nourin inhibitors/antagonists are aimed at controlling the level and activity of Nourin, they will likely reduce early post-reperfusion leukocyte influx/activation and inflammation-induced damage without interfering with the crucial cardiac repair process that occurs a few days following myocardial infarction [10]. Additionally, an independent study indicated that high levels of TNF-α, IL-1β, and IL-8 were detected in UA patients during the early disease phase [16], suggesting a role of Nourin-induced proinflammatory cytokines in the pathogenesis of UA. Clinically, using Nourin ELISA and chemotaxis assay, we demonstrated that high levels of Nourin were detected in blood samples collected at presentation to hospital EDs from symptomatic acute coronary syndromes (ACS) patients and patients with documented AMI but not in symptomatic noncardiac patients and healthy subjects [17]. These results suggest the potential use of the Nourin biomarker to rule out myocardial ischemia for symptomatic patients having noncardiac causes.

Non-coding RNA molecules are nonprotein coding genes that regulate 60% of protein coding genes. Non-coding RNAs are either: (1) short noncoding of approximately 22–24 nucleotides, named miRNAs post-transcriptional regulator, or (2) long noncoding (>200 nucleotides) [18]. They are involved in numerous biological processes, including cell proliferation, differentiation, and apoptosis, thereby contributing to various pathological conditions [19]. With the development of high-throughput next-generation sequencing (NGS), about 30 miRNAs have been reported as specific diagnostic and prognostic blood-based biomarkers in AMI patients, including *miR-155*, *miR-380*, *miR-208b*, and *miR-133a.* On the other hand, a cluster of the three miRNAs—(*miR-150*/*186/132*), miR-133a, and *miR-108b*—are linked to UA patients [20,21]. The role of miR-137 in ischemic stroke and their potential therapeutic targets have also been reported in recent studies [22,23]. Although the focus was restricted to cerebral [22] and retinal hypoxia [24], the signaling pathway was directly related to hypoxic conditions. Li et al. reported that *miR-137* is a hypoxia-responsive gene, and its overexpression aggravates hypoxia-induced cell apoptosis [24]. Additionally, recent bioinformatics analysis has suggested that *miR-106b-5p* serves as a robust inflammatory mediator for apoptosis and angiogenesis in AMI [25,26,27].

In the current study, we applied an integrative bioinformatics analysis using the Nourin amino acid sequence to retrieve the Nourin-associated miRNAs in UA and acute STEMI patients. The two miRNAs, *miR-137* and *miR-106b-5p*, which are linked to myocardial ischemia, were selected. Both miRNAs are found to be expressed in human hearts and they were identified as cardiac-enriched miRNAs that are found to be overexpressed at a much higher level during cardiac tissue injury and in AMI patients, but undetectable in healthy individuals. Furthermore, the lncRNA/miRNA/mRNA network for each miRNA was constructed and a quantitative real time PCR (qPCR) was performed to determine whether the Nourin-dependent miRNAs, *miR-137* and *miR-106b-5p*, can diagnose ischemia-induced injury in UA patients when troponin levels are below the clinical decision limit or in STEMI patients with elevated troponin level. A schematic diagram illustrating the signaling pathway of Nourin-dependent *miR-137* and *miR-106b-5p* in myocardial ischemic injury is presented in Figure 2. Specifically, downregulation of *lncR-CTB89H12.4* due to ischemia resulted in upregulation of both miR-137 (marker of cell damage) and miR-106b-5p (marker of inflammation), as well as activation of *mRNA-FTHL-17* and *mRNA-ANAPC11* with a likely increased translation and production of Nourin protein. Thus, the association of these miRNAs with their regulatory molecular networks outlines a novel underlying mechanistic signaling pathway in myocardial ischemic injury.

## 2. Subjects and Methods

### 2.1. Bioinformatics Analysis

A comprehensive bioinformatics analysis was conducted to analyze different expressed genes (DEGs) that are incorporated with myocardial ischemic injury and related to Nourin protein, then the Nourin protein interaction was mapped through functional gathering analysis using the incorporated gene ontology (GO) and Kyoto Encyclopedia of Genes and Genomes (KEGG) databases, and finally, a pathway enrichment analysis of the DEGs was performed using the online The Database for Annotation, Visualization and Integrated Discovery (DAVID ) tools. A *p*-value < 0.05 indicates significance. To construct a protein–protein interaction (PPI) network, the Nourin-dependent DEGs were uploaded to an interacting genes/proteins database (*STRING*), which provides a systematic functional organization of the proteome. The data obtained revealed that ferritin heavy chain like 17 (*FTHL-17*) and anaphase promoting complex subunit 11 (*ANAPC11*) genes directly targeted Nourin protein, and they were significantly dysregulated in myocardial ischemic injury. In addition, they were minimally expressed in normal myocardial tissue. To construct lncRNA/miRNA/mRNA networks for the two selected genes, these target genes were further analyzed to predict specific miRNAs, using miRDB, TargetScan and miRTarBase databases. Analysis showed that *miR-137* and *miR-106b-5p* are candidate targets for *FTHL-17* and *ANAPC11*, respectively. Each mRNA was recognized by three different databases to ensure that data accuracy and selection were based on detection of high complementarity binding sites. Finally, the *lncRNA-CTB89H12.4* (*CTB89H12.4*) was selected to control the expression of both miRNAs using the DIANA-LncBasev3, Starbase and CHIP databases [28]. The in silico prediction of each network was verified by Clustal 2 multiple sequence alignment software to validate the alignment between the candidate genes in each network.

### 2.2. Study Design/Participant’s Selection

A prospective observational study was conducted on UA and acute STEMI patients presented with chest pain at Kasr-Alainy hospital during the period from October 2019 to March 2020. The study complied with the Declaration of Helsinki and was approved by the Ethics Community of Faculty of Medicine, Cairo University. A written consent was obtained from each patient and healthy subject before enrollment.

In the current pilot study, all 62 subjects met the inclusion/exclusion criteria before being enrolled, including UA (*n* = 30) and STEMI (*n* = 16) patients with acute chest pain within the first 10 h of symptoms, as well as healthy subjects (*n* = 16). Unstable angina patients were defined as patients with prolonged angina pain at rest, new onset angina, or recent destabilization of previously stable angina. Diagnosis of UA patients was conducted using (1) high-sensitive cardiac troponin T (hs-cTnT) (Elecsys Troponin T hs, and some Troponin I, Roche Diagnostics) and (2) invasive coronary angiography. Acute STEMI patients were diagnosed using ECG ST-segment changes and troponin levels. Healthy participants (*n* = 16) underwent standard clinical physical evaluation, measured troponin levels and exercised on an ECG/treadmill stress test for 10 min to confirm absence of myocardial ischemia. The following patients were excluded from the study: patients with recent attack of AMI, positive c-reactive protein (CRP), cardiomyopathy, congenital heart disease, heart failure, renal failure, hepatitis, hepatic failure, bleeding disorders, malignancy, and autoimmune diseases, including arthritis and inflammatory bowel disease.

### 2.3. Total and miRNA Extraction and Purification

Venous blood (3 mL) was collected at zero-time from patients at presentation to the hospital ED with acute chest pain; blood was centrifuged at 1300× *g* at 4 °C for 20 min, and the serum was immediately stored at −80 °C until analyzed. In a limited study, venous blood was also collected, centrifuged at 1300× *g* at 4 °C for 20 min, and the plasma (EDTA) was immediately stored at −80 °C until analyzed. Total RNA and miRNAs were extracted from sera samples using the RNeasy Mini Kit (Qiagen, Hilden, Germany) according to the manufacturer’s protocol, and the concentration and purity of RNA were evaluated spectrophotometrically at 260 and 280 nm, respectively. RNA integrity was visually confirmed by agarose gel electrophoresis. Using miScript RT Kit (Qiagen, Hilden, Germany), cDNA was synthesized by reverse transcription reaction.

### 2.4. Gene Amplification Analysis by Quantitative Real Time PCR (qPCR)

cDNA was amplified for miRNA, mRNA, and lncRNA expression using miScript primer assay for miRNA amplification (Hs_*miR-137*_1 and Hs_*miR-106b-5p_1* miScript primer assay) and Quantitect primer assays for mRNAs (Hs_*FTHL17_1* and Hs_*ANAPC11_1*_SG QuantiTect primer assays) and lncRNA (RT^2^_*CTB89H12.4* primer assay). The Hs_*RNU6-2_11* and Hs_*GAPDH* genes were used as reference housekeeper genes. The thermal cycling was adjusted according to manufacture instructions. The PCR analysis was conducted on Rotor-Gene Q 5plex HRM Platform (Qiagen, Hilden, Germany). The fluorescence data were collected at the extension step. Following amplification, gene expression was calculated using the 2^∆∆Ct^ method.

### 2.5. Statistical Analysis

Statistical analysis was performed using GraphPad Prism software, version no: 8.0.2. Calculation of median and range was used for skewed data, while calculation of mean ± SD was used for normally distributed numerical data. Calculation of number of cases and percentages used cross-tabulation. The nonparametric Mann–Whitney U test was used to compare the expression of genes between two groups, and ANOVA was used to evaluate the difference in gene expression for more than two groups. The diagnostic potential of miRNAs for UA and acute STEMI was evaluated by receiver operator characteristics (ROC) analysis, then the biomarker sensitivities and specificities were calculated according to the optimum cut-off value. Significance was set at *p* ≤ 0.05.

## 3. Results

### 3.1. Demographic Characteristics of Participants

The clinical characteristics of patients with UA (*n* = 30), patients with acute STEMI (*n* = 16) and 16 healthy controls are presented in Table 1.

### 3.2. Assessment of CTB89H12.4/miR-137/FTHL-17 Network

The expression of candidate genes was verified using qPCR. For *miR-137* expression (Figure 3a), there was significant upregulation by 1382-fold (*p* = 0.001) in UA patients compared to healthy control, while *FTHL-17* expression (Figure 3b) had a 1.7-fold increase in UA patients compared to healthy. However, the *CTB89H12.4* gene was downregulated by 20-fold in UA compared to healthy (*p* = 0.0001) and by 3.4-fold in acute STEMI compared to UA (*p* = 0.001) (Figure 3c) (Table 2). Interestingly, comparing gene expression of *miR-137* and *FTHL-17* in UA to acute STEMI patients revealed that *miR-137* (Figure 3a) showed a significant increase by 2.5-fold (*p* < 0.01) in patients with STEMI compared to UA and that *FTHL-17* (Figure 3b) also showed a significant increase by 7.5-fold (*p* < 0.0001) in patients with acute STEMI compared to UA. Additionally, the expression pattern for miR-137 in UA and acute STEMI patients was plotted in an XY graph to illustrate the marked elevated expression levels in UA and STEMI patients compared to the observed minimal expression in healthy controls (Figure 3f).

Furthermore, an ROC curve analysis was conducted to evaluate the reliability of miR-137 as a diagnostic biomarker for UA and STEMI patients. Comparing UA to healthy controls, the calculated cut-off value was 195.4 (Figure 4a). The *miR-137* biomarker sensitivity and specificity were 97% and 94%, respectively, to discriminate UA from healthy controls (area under the curve (AUC): 0.99, Youden’s index: 0.91, *J* value = 195.4). Additionally, at an optimum cut-off value of 2488, miR-137 expression level could potentially discriminate UA from patients with acute STEMI (AUC: 0.84, Youden’s index: 0.58, *J* value = 157) (Figure 4b) with sensitivity of 75% and specificity of 83%.

### 3.3. Assessment of CTB89H12.4/miR-106b-5p/ANAPC11 Network

Serum *miR-106b-5p* gene expression was associated with myocardial ischemia, where *miR-106b-5p* (Figure 3d) and ANAPC11 (Figure 3e) were significantly (*p* = 0.001) upregulated by 192-fold and 8.5-fold in UA patients compared to healthy subjects, respectively. Comparing gene expression in UA with STEMI patients, *miR-106b-5p* (Figure 3d) expression was significantly increased by 4.6-fold (*p* < 0.001) in patients with acute STEMI compared to UA, but there was no statistically significant difference in *ANAPC11* gene expression between UA and STEMI (Figure 3e). The *CTB89H12.4* gene was downregulated by 20-fold in UA compared to healthy (*p* = 0.0001) and by 3.4-fold in STEMI compared to UA (*p* = 0.001) (Figure 3c) (Table 2). In addition, the expression pattern for *miR-106b-5p* in UA and STEMI patients was plotted in an XY graph to illustrate the sizable difference in expression level in UA and STEMI compared to the observed minimal expression in healthy controls (Figure 3g). The diagnostic potential of *miR-106b-5p* was assessed and the ROC analysis revealed that *miR-106b-5p* is a reliable biomarker for UA and STEMI. The calculated biomarker sensitivities and specificities were 87% and 88%, respectively, to discriminate UA from healthy controls; the optimum cut-off value was 90.4 (AUC: 0.9, Youden’s index: 0.75, *J* value: 157) (Figure 4a). To discriminate UA from STEMI patients (Figure 4b), results indicated that at a cut-off value of 385.0 (AUC: 0.95, Youden’s index: 0.76, *J* value: 175), the diagnostic sensitivity was 86% and specificity was 90%.

There was no gender difference in *miR-137* (Figure 4c) and *miR-106b-5p* (Figure 4d) gene expression level in STEMI patients (*p* > 0.05). However, there was higher expression of *miR-137* in males UA compared to females (*p* < 0.05) (Figure 4c) but no gender difference in *miR-106b-5p* gene expression in UA patients (Figure 4d). Additionally, in a limited study (*n* = 4 for each group), the gene expression levels of *miR-137* and *miR-106b-5p* were measured in plasma samples and compared to their levels in serum samples. There was no significant difference (*p* > 0.05) in *miR-137* (Figure 4e) and *miR-106b-5p* (Figure 4f) gene expression detected between serum and plasma samples collected from UA, STEMI, and healthy.

Troponin analysis indicated that during serial sampling at the 0/3 h, 67% of UA patients had hs-cTnT levels consistently below the 99th percentile value (14 ng/L), while only 33% of patients showed low-to-mild elevation of hs-cTnT levels above the 99th percentile values within the range of 15 ng/L to 51 ng/L but without significant delta change in serial samples. Unstable angina was further confirmed by invasive coronary angiography, where more than 50% of UA patients had coronary artery stenosis. Contrary to the limited elevation of hs-cTnT levels of <1 to 3-fold over the cutoff of 14 ng/L in only 33% of UA patients, Nourin miRNAs were markedly elevated in 100% of UA patients, with a significant increase of 1382-fold for *miR-137* and 192-fold for *miR-106b-5p* over baseline gene expression levels in healthy subjects. For acute STEMI patients, myocardial infarction was confirmed by persistent ST-segment changes and significant high elevation of hs-cTnT in all patients, similar to the marked elevation of Nourin miRNAs in the same 16 STEMI patients compared to healthy (1535-fold increase for *miR-137* and 177-fold for *miR-106b-5p*). Healthy participants represented the negative control group, and their troponin levels were below the clinical decision limit and showed negative ECG/treadmill stress tests, indicative of absence of myocardial ischemia. Healthy subjects also showed minimum gene expression of Nourin miRNAs.

### 3.4. Correlation between lncRNA/miRNA/mRNA Genes in Acute Coronary Syndrome (ACS) Patients

Since there was no significant association detected between the expression of *miR-137*, *FTHL-17*, and *CTB89H12.4*, as well as for *miR-106b-5p*, *ANAPC11*, and *CTB89H12.4* genes for UA or patients with STEMI, we combined the two groups (UA and acute STEMI) into one group called ACS group, and then a Spearman’s analysis was conducted between the two paired genes in each network. Results revealed a positive significant correlation between *miR-137* and *FTHL-17* (*r*: 0.5, 95% CI: 0.2–0.7, *p* = 0.001) and between *miR-106b-5p* and *ANAPC11* (*r*: 0.4, 95% CI: 0.4–0.6, *p* = 0.02) genes in ACS patients. In contrary, *CTB89H12.4* showed a significant negative correlation with either *FTHL-17* or *ANAPC11*. Lower expression levels of *CTB89H12.4* were significantly associated with higher expression levels of *FTHL-17* and *ANAPC11* in ACS patients (*p* < 0.01).

In summary, Nourin *miR-137* (marker of cell damage) and *miR-106b-5p* (marker of inflammation) gene expression measured in serum samples collected from patients with acute chest pain at presentation to the hospital ED, could (1) detect unstable angina patients with chest pain secondary to reversible myocardial ischemia and confirmed by invasive coronary angiography to have greater than 50% coronary artery stenosis and troponin level below the clinical decision, and (2) detect acute STEMI patients with chest pain secondary to irreversible myocardial ischemia (necrosis) and confirmed by persistent ST-segment changes and elevated troponin level. Additionally, both biomarkers showed consistent elevation of Nourin *miR-137* and *miR-106b* gene expression in all 30 UA patients and 16 acute STEMI patients. Therefore, Figure 2 presents an underlying molecular mechanism associated with the Nourin protein and its RNA molecular network in UA and acute STEMI patients. The downregulation of *CTB89H12.4* after myocardial ischemia results in an increase in the expression of miR-137 and miR-106b-5p, which sequentially activates the *FTHL-17* and *ANAPC11*, respectively, with an increased translation and production of Nourin protein. Results also support the ontology bioinformatics evidence that *CTB89H12.4*/*miR-137*/*FTHL-17* and *CTB89H12.4*/*miR-106b-5p*/*ANAPC11* networks synergistically regulate the Nourin protein expression in myocardial ischemia, and thus provide a novel molecular mechanism in ischemic heart disease.

## 4. Discussion

Acute coronary syndrome is a life threatening condition owing to high morbidity and mortality worldwide, due partially to the lack of early blood-based biomarkers that can diagnose reversible myocardial ischemia before progressing to necrosis, and thus can guide treatment decisions [6]. Although high sensitivity troponin represents the gold standard cardiac specific biomarker [29], the major limitation is its inability to diagnose unstable angina patients [7], and its low specificity in differentiating ACS patients from other nonischemic clinical emergency syndromes, such as acute myocarditis, heart failure without CAD, stress-induced cardiomyopathy, and pulmonary embolism [30]. Accordingly, there is an ultimate need to identify novel early and sensitive biomarkers to specifically diagnose unstable angina patients and assist in their therapeutic approaches.

Nourin is a novel blood biomarker with a known mode of action as an early potent inflammatory mediator released within 5 min by ischemic myocardial tissue and it is associated with the initiation and amplification of post-ischemic cardiac inflammation [10,11,12,13,14,15]. Clinically high levels of Nourin were detected in ACS and AMI patients, but not in symptomatic noncardiac patients and healthy subjects [17], suggesting a potential use of the Nourin biomarker to rule out myocardial ischemia for symptomatic patients having noncardiac causes. Recently, aberrant miRNAs have been reported to play a role in the pathogenesis of cardiovascular diseases, and they are involved in vascular dysfunction, apoptosis, autophagy, inflammation, and angiogenesis [9,18,31,32,33]. In the present study, we applied a bioinformatics analysis to elucidate the underlying molecular mechanism associated with Nourin protein, and accordingly, the topological characteristics between Nourin protein sequence and miRNAs-mediated interactions network associated with myocardial ischemia were retrieved. The two networks, *CTB89H12.4*/*miR-137*/*FTHL-17* and *CTB89H12.4*/*miR-106b-5p*/*ANAPC11*, were selected based on novelty, their direct interaction with Nourin protein, as well as association with myocardial ischemia. Both networks showed high tendency to interact with each other through sharing in the lncRNA *CTB89H12.4*. Standard qPCR was used in order to evaluate the diagnostic efficacy of Nourin-dependent miRNAs (*miR-137* as a marker of cell damage and *miR-106b-5p* as a marker of inflammation) for ruling in patients with UA when troponin level was below the clinical decision limit for diagnosis and patients had greater than 50% coronary artery stenosis by invasive coronary angiography. Gene expression profiles of circulating *miR-137* and *miR-106b-5p* and the association with their target genes *FTHL-17*, *ANAPC11*, and *CTB89H112.4* were investigated in patients with UA (*n* = 30), acute STEMI (*n* = 16) and healthy subjects (*n* = 16). Results indicate that *miR-137* and *miR-106b-5p* were significantly upregulated in UA and acute STEMI patients compared to healthy controls and that both miRNAs can diagnose symptomatic UA patients with high sensitivity and specificity, as well as stratify severity of myocardial ischemia with higher expression in acute STEMI compared to UA. Furthermore, the significant association of *miR-137* and *miR-106b-5p* with their regulatory target genes *FTHL-17*, *ANAPC11*, and *CTB89H12.4* supported the bioinformatics data and outlined a novel underlying mechanistic signaling pathway in myocardial ischemic injury. However, the small sample size reflects the limitation of the study and, thus, a large-scale sample size study is needed with focus on the Nourin underlying mechanistic pathways as novel cardiac biomarkers.

Recent evidence has demonstrated that *miR-137* is a hypoxia-responsive gene over-expressed in ischemic-cell injury [24]. Moreover, recent studies have reported that downregulation of *miR-137* inhibits oxidative stress-induced cardiomyocytes apoptosis by targeting the *KLF15* gene [34] and protects cells against ischemia-induced apoptosis through *CDC42* [35,36]. On the other hand, high expression levels of *miR-137* have been observed in dexamethasone-treated multiple myeloma cells by targeting *MCL-1* and *MITF* genes [37,38], thus inhibiting cancer cell proliferation and inducing apoptosis [39,40]. These findings suggest a regulatory mechanistic pathway of *miR-137* in ischemia-induced cell injury, which has been investigated in neural cells on wide scale [22]. Recently, the therapeutic value of *miR-137* has been reported in hypoxia-induced retinal diseases, through targeting the Notch1 signaling pathway [41].

In addition to the reported inflammatory mechanism of *miR-106b-5p* in myocardial tissue, it was described as a hypoxia-associated miRNA in myocardial infarction [27], as well as various human cancers [42,43]. *MiR-106b-5p* exerts cell degradation through a lysosomal-dependent mechanism. Upregulation of *miR-106b-5p* has been demonstrated in myocardial tissue with dilated cardiomyopathy in which the autophagy mechanism is mediated through the *miR-106b-5p*/*FYCO1*-dependent pathway [44]. Additionally, *miR-106b-5p* was found to be the most significant miRNA in 746 altered miRNAs in atherosclerotic vascular diseases; it targets multiple inflammatory signaling pathways including PI3K, Akt, mTOR, TGF-β, TNF, TLR, and HIF-1α [44]. In vitro experimental studies have reported its biological role in cholesterol homeostasis [45] and spermatogenesis [46]. The regulatory role of *miR-106b-5p* in cardiomyocytes apoptosis has been investigated to illustrate the underlying signaling pathway in myocardial infarction [27]. Recent evidence suggested that suppression of *miR-106b-5p* promotes hypoxia-induced cell apoptosis, an observation that explains the early increase in its serum level in response to ischemia-induced myocardial injury. As miRNAs are tissue specific, a previous study suggested that p21 is the direct target of *miR-106b-5p* in cardiomyocytes that mediate an anti-apoptotic effect under hypoxic conditions [27]. It has been established that *miR-137* and *miR-106b* are strongly linked to the pathogenesis of atherosclerosis [47]. Recently, *miR-137* was shown to directly suppress the insulin-like growth factor-binding protein 5 (IGFBP-5), which further suppresses cell proliferation and migration of vascular smooth muscle cells (VSMCs) [48]. Additionally, overexpression of *miR-137* in VSMCs suppresses the activity of mTOR/STAT3 signaling, and *miR-106b* modulates angiogenesis in vascular endothelial cells by targeting STAT3 [49].

As reported here, a significant positive correlation was detected in UA and acute STEMI patients between *miR-137* and *FTHL-17*, as well as between *miR-106b-5p* and *ANAPC11*. However, a significant negative correlation was detected between the two miRNAs (*miR-137* and *miR-106b-5p*) and *lnc-CTB89H12.4*. In the context of identifying new cardiac diagnostic biomarkers with high sensitivity and specificity, it is more beneficial to use a network of functionally linked biomarkers such as the Nourin RNA molecular network rather than depending on a single biomarker. Specifically, the Nourin-dependent *FTHL-17* gene is a novel type of ferritin gene, located on X chromosome. It encodes for ferritin-like protein but without ferroxidase activity. The partial nuclear localization of *FTHL-17* suggests its specific biological functions apart from iron storage regulation [50]. Similarly, the Nourin-dependent *ANAPC11* genes are a complex of subunits that promote cell cycle progression and induce G1 arrest through ubiquitination-mediated cell destruction and have two novel insights, including protein degradation and epigenetic regulation [51]. Based on gene database records, the two genes (*FTHL-17* and *ANAPC11*) and their miRNAs regulators (*miR-137* and *miR-106b-5p*) are highly expressed in cardiomyocytes and skeletal muscles, which strongly support their symbiotic interaction. Additionally, the role of *lnc-CTB89H12.4*, also called *lncRNA-RP11-175K6.1*, has been reported in atherosclerosis and angiogenesis, which highlights its strong association with angiogenesis and cardiomyocytes [52,53].

The study has points of strength and some limitations. The relatively small sample size of patients at zero time, as well as the limited baseline miRNA measurements, represent the first limiting point. The second point is that despite the increased quality of the study design by accurate diagnosis of patients through elective coronary angiography, the study population does not represent the general population, since all cardiac patients were enrolled from a single hospital, Kasr-Alainy Hospital. In addition, we used a combination of two Nourin-dependent miRNAs biomarkers measured with highly sensitive PCR technique, which enabled us to avoid bias created by a raised number of false positive results. Finally, the study showed restricted interpretation of results due to lack of long-term follow-up and correlation with severity of myocardial injury.

## 5. Conclusions

Despite availability of high-sensitivity cardiac troponin, there is a need for new biomarkers with high diagnostic sensitivity and specificity for the detection of myocardial ischemia in suspected ACS patients with troponin levels below the clinical decision limit. We showed that the Nourin-dependent *miR-137* and *miR-106b-5p* are a novel promising rule-in test that can (1) diagnose unstable angina among patients presenting to hospital EDs with acute chest pain, and (2) stratify severity of myocardial ischemia, with higher expression in acute STEMI compared to UA patients. The diagnostic potential of *miR-137* and *miR-106b-5p* indicated that both miRNAs are reliable biomarkers for UA and acute STEMI patients. Finally, early diagnosis of suspected unstable angina patients using the novel Nourin biomarkers is key for initiating guideline-based therapy that improves patients’ health outcomes.

## Figures and Tables

**Figure 1 biomolecules-11-00368-f001:**
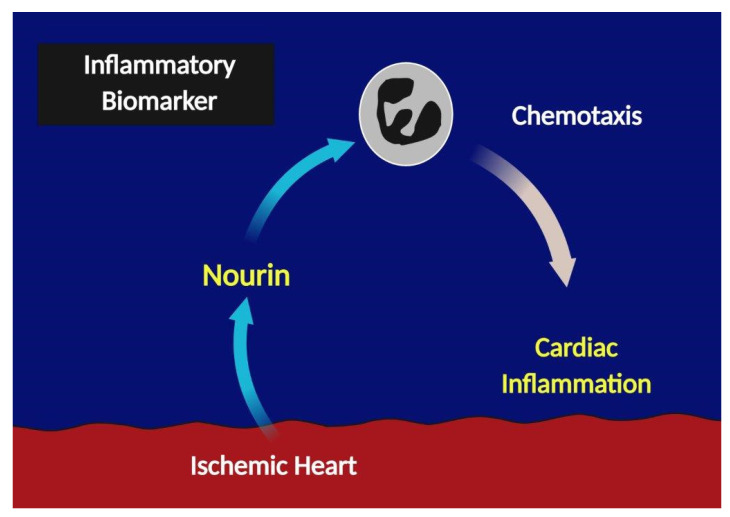
Schematic diagram illustrating the release of the leukocyte chemotactic factor Nourin by ischemic myocardium, followed by leukocyte recruitment and cardiac inflammation.

**Figure 2 biomolecules-11-00368-f002:**
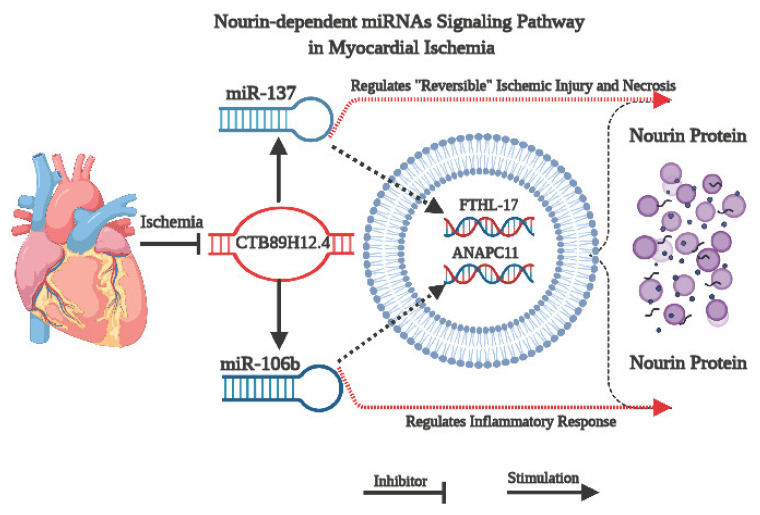
Schematic diagram illustrating the mechanism of Nourin-dependent miRNAs, *miR-137* and *miR-106b-5p*, in myocardial ischemic injury and their associated molecular network, based on integrative bioinformatics analysis. Downregulation of *CTB89H12.4* due to ischemia resulted in upregulation of both *miR-137* (marker of cell damage) and *miR-106b-5p* (marker of inflammation) and activation of *mRNA-FTHL-17* and *mRNA-ANAPC11*, with a likely increased translation and production of Nourin protein.

**Figure 3 biomolecules-11-00368-f003:**
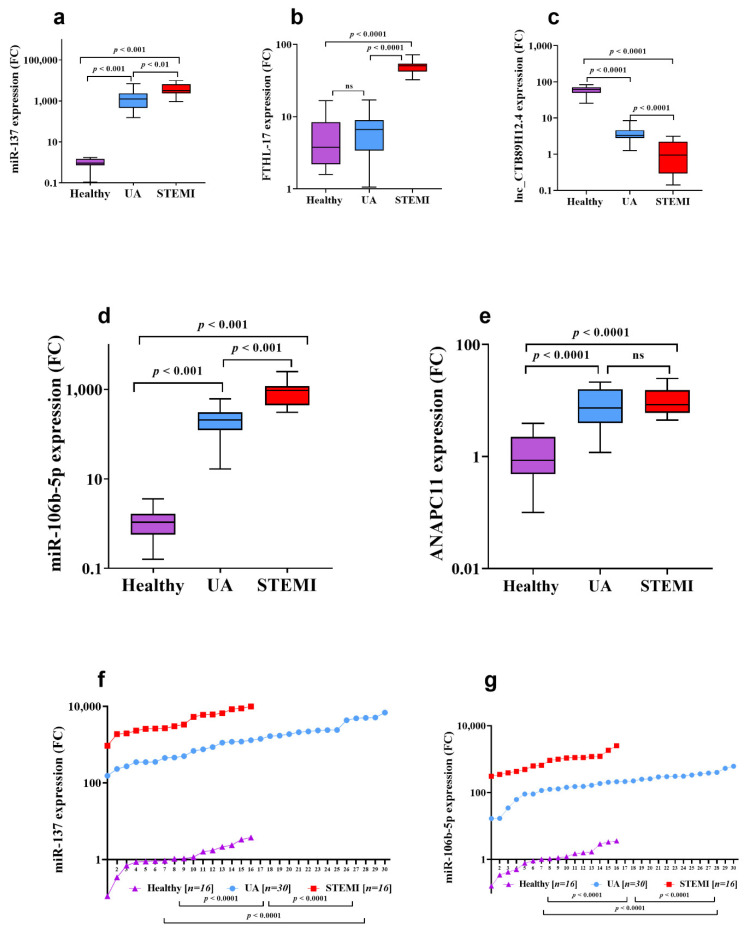
Expression profiles of candidate miRNAs in patients with UA and acute STEMI in comparison to healthy controls. The two candidate miRNAs, *miR-137* and *miR-106b-5p*, were measured with qRT-PCR using 62 serum samples, including 16 Healthy (non-ACS) as well as 30 UA and 16 STEMI patients (ACS samples). *miR-137* (**a**) and *miR-106b-5p* (**d**) were significantly upregulated in the UA and STEMI groups compared to Healthy (**a**,**d**). In addition, similar upregulation was observed for the associated mRNA targets *FTHL-17* (**b**) and *ANAPC11* (**e**), while the *lnc-CTB89H12.4* regulating gene was significantly decreased in UA and STEMI patients (**c**). (**f**,**g**) demonstrate the expression pattern of *miR-137* and *miR-106b-5p* in the three studied groups (UA, STEMI and Healthy). Abbreviation: ns: no significant difference.

**Figure 4 biomolecules-11-00368-f004:**
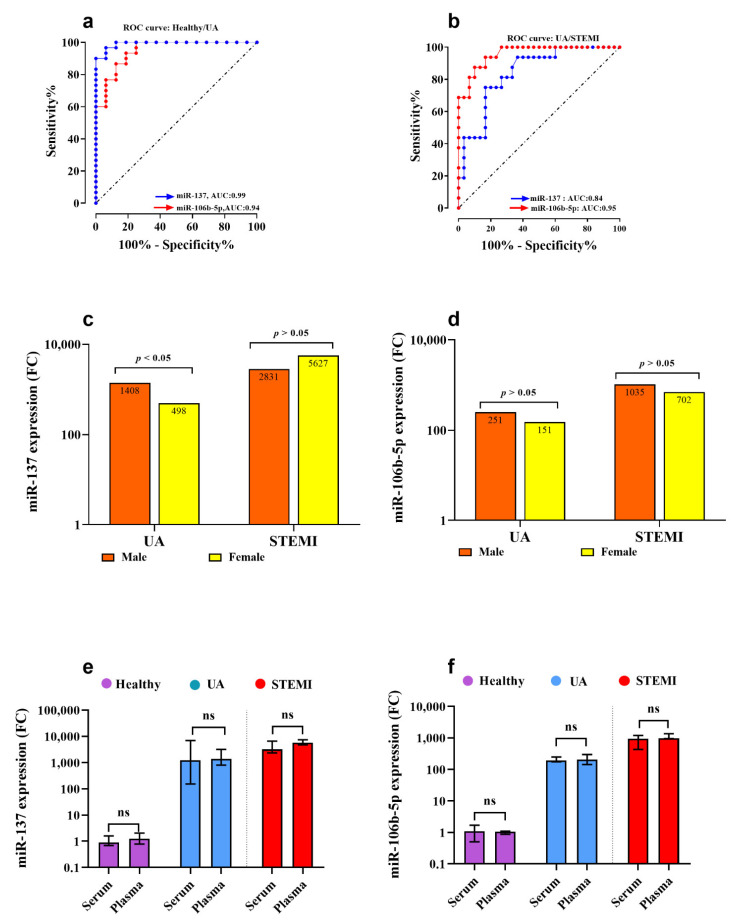
Receiver operator characteristics (ROC) curve illustrating the diagnostic value of the two miRNAs (*miR-137* and *miR-106b-5p*) in discriminating UA from Healthy (**a**) and UA from STEMI (**b**). (**c**,**d**) show the nonsignificant association of miR-106b-5p expression level between genders in UA and AMI patients, respectively. Finally, the serum and plasma expression levels for *miR-137* (**e**) and *miR-106b-5p* (**f**) were compared, and results indicate no significant difference was reached. Abbreviations: UA: unstable angina, STEMI: ST-elevation myocardial infarction, AUC: area under the curve, *p*-value > 0.05 considered statistically insignificant, and *p*-value < 0.05 considered statistically significant.

**Table 1 biomolecules-11-00368-t001:** Clinical Characteristics of Patient Populations.

Variables	ACS (*n* = 46)		Healthy (*n* = 16)	*p*-Value
	UA (*n* = 30)	STEMI (*n* = 16)		
Age (years) mean ± SD	60.1 ± 8.1	54.4 ± 12.7	32.9 ± 9.9	0.001
Sex: Males: *n* (%)	19 (63.3)	12 (75)	16 (100)	0.014
Risk factors				
BMI (kg/m^2^) mean ± SD	29.3 ± 7.1	31.4 ± 4.8	26.8 ± 4.7 8 (50)	0.145
Smoking: *n* (%)	16 (53.3)	10 (52.5)		0.09
Diabetes Mellitus: *n* (%)	16 (53.3)	6 (37.5)		0.001
Hypertension: *n* (%)	19 (63.3)	7 (43)		0.001
Dyslipidemia: *n* (%)	12 (40)	5 (31.3)		0.001

ASC: Acute coronary syndrome, UA: Unstable angina, STEMI: ST-segment elevated myocardial infarction, BMI: Body mass index, *n*: Number of cases, %: Percentage of cases calculated as percentage in the group, SD: Standard deviation, *p*-value > 0.05 considered statistically insignificant, and *p*-value < 0.05 considered statistically significant.

**Table 2 biomolecules-11-00368-t002:** Gene Expression Profiles in Patients with UA versus Healthy, as well as Patients with UA versus Acute STEMI.

Genes	Median Expression Level (log_10_)			Statistics FC (*p*-Value)	
	Healthy	UA	STEMI	UA/Healthy	STEMI/UA
***miR-137*** (log_10_)	0.9	1244	3163	1382 (<0.0001)	2.5 (<0.001)
***FTHL-17*** (log_10_)	3.76	6.6	50.4	1.7 (<0.0001)	7.5 (<0.0001)
***miR-106b-5p*** (log_10_)	1.08	207	953	192 (<0.001)	4.6 (<0.001)
***ANAPC11*** (log_10_)	0.86	7.36	8.39	8.5 (<0.0001)	1.1 (>0.05)
***CTB89H12.4*** (log_10_)	65.3	3.27	0.94	20 (<0.0001)	3.4 (<0.001)

UA: Unstable angina, STEMI: ST-segment elevated myocardial infarction, FC: Fold change, *FTHL-17*, ferritin heavy chain like 17, *ANAPC11*: Anaphase promoting complex subunit, *p*-value > 0.05 considered statistically insignificant, and *p*-value < 0.05 considered statistically significant.

## Data Availability

The data presented in this study are available on request from the corresponding author. The data presented in this study are available on request from the corresponding author. The data are not publicly due to privacy.

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
