# Peer review of "Nourin-Dependent miR-137 and miR-106b: Novel Early Inflammatory Diagnostic Biomarkers for Unstable Angina Patients"

_biomolecules, 2021, doi:10.3390/biom11030368_

Round 1

Reviewer 1 Report

What is the source of these miRNAs? Which cell type?

The strengths and limitations of the study should be deeply addressed, taking into account sources of potential bias or imprecision: Discuss both direction and magnitude of any potential bias.

The role of miR-137 and miR-106b in atherosclerosis (PMID: 29016699, PMID: 26662987) should be better discussed.

Please avoid the use of red and green (issues in color blind Readers) in the same figure.

Author Response

Subject: Revision on a manuscript entitled “Nourin-Dependent miR-137 and miR-106b:  Novel Early Inflammatory Diagnostic Biomarkers for Unstable Angina Patients”

Thank you for your e-mail regarding our manuscript and we welcome the opportunity to revise it according to both the editors and the reviewers’ comments.  We greatly appreciate the time and attention given in reviewing our manuscript and we thank the reviewers for their helpful suggestions. Each point raised has been carefully addressed, and our responses are detailed point-by-point below. Revisions made to the manuscript text, in response to the comments, have been marked using track changes.

Reviewer 1

 -Reviewer Comment 1: What is the source of these miRNAs? Which cell type?

Author response and action taken:

Through Bioinformatics analysis “Genes/Proteins interactions” using the Nourin amino acid sequence, were found that miR-137 and miR-106b to be absolutely linked to the Nourin protein expressed as an early and direct response to myocardial ischemia. Moreover, both miRNAs are found to be expressed in “human” hearts. They are both identified as cardiac-enriched miRNAs that found to be overexpressed in much higher level during cardiac tissue injury and AMI patients. In addition, miR-137 and miR-106b are undetectable in healthy individuals.

This clarification is now included in page 7 of the revised manuscript and marked by track changes.

-Reviewer Comment 2: The strengths and limitations of the study should be deeply addressed, taking into account sources of potential bias or imprecision: Discuss both direction and magnitude of any potential bias.

Author response and action taken:

We agree with the editor comments; therefore, we have added a paragraph at the end of the discussion section, in which we highlighted the magnitude of both direction of bias. The added paragraph is in page 25 of the revised manuscript and marked by track changes.

The edited paragraph is The study has points of strength and some limitations. The relatively small sample size of patients at zero time, as well as the limited baseline miRNA measurements, represent the first limiting point. The second point is that the study population doesn’t represent the general population, since all cardiac patients were enrolled from a single hospital “Kasr-Alainy hospital “, despite the increased quality of the study design by accurate diagnosis of patients through elective coronary angiography. In addition, we used a combination of two Nourin-dependent miRNAs biomarkers measured with highly sensitive PCR technique, which enabled us to avoid bias created by raised number of false-positive results. Finally, the study showed restricted interpretation of results due to lack of long-term follow-up and correlation with severity of myocardial injury”

-Reviewer Comment 3: The role of miR-137 and miR-106b in atherosclerosis (PMID: 29016699, PMID: 26662987) should be better discussed.

Author response and action taken:

Thank you for your constructive comment. We are limited by the number of word limit of the journal and we tried to be strictly specific to the research objectives which is concerned with the Nourin dependent miRNAs “miR-137 and miR-106b-5p” in relation to ischemic injury. However, we do agree with the reviewer suggestion which will raise the value of the manuscript and will be interesting for the reader to add more knowledge about the role of both miRNAs in atherosclerosis. Therefore, we give a brief highlight on the role of miR-137 and miR-106b in atherosclerosis and three new references are added and cited in the revised manuscript. The below added paragraph is in page 24 of the revised manuscript and marked by track changes.

"It has been established that miR-137 and miR-106b are strongly linked to the pathogenesis of atherosclerosis [47]. Recently, miR-137 was shown to directly suppress the Insulin Like Growth Factor Binding Protein 5 (IGFBP-5), which further suppresses cell proliferation and migration of Vascular Smooth Muscle Cells (VSMCs) [48]. Additionally, overexpression of miR-137 in VSMCs suppresses the activity of mTOR/STAT3 signaling, and that miR-106b modulates angiogenesis in vascular endothelial cells by targeting STAT3 [49].

-Reviewer Comment 4: Please avoid the use of red and green (issues in color blind Readers) in the same figure.

Author response and action taken:

We thank the reviewer for the valuable comments and we have changed the green colour in all figures and replaced it with another colour which could be easily distinguishable from red colour.

Reviewer 2 Report

The work is interesting and has practical application in medicine, so I think it is worth publishing.

Minor errors:

Fig2. There should be "regulates" instead of "reguates".

Fig3. Figures are too small, legend is not vosible.

Author Response

Subject: Revision on a manuscript entitled “Nourin-Dependent miR-137 and miR-106b:  Novel Early Inflammatory Diagnostic Biomarkers for Unstable Angina Patients”

Thank you for your e-mail regarding our manuscript and we welcome the opportunity to revise it according to both the editors and the reviewers’ comments.  We greatly appreciate the time and attention given in reviewing our manuscript and we thank the reviewers for their helpful suggestions. Each point raised has been carefully addressed, and our responses are detailed point-by-point below. Revisions made to the manuscript text, in response to the comments, have been marked using track changes.

Reviewer # 2

-Reviewer Comment 1. – The work is interesting and has practical application in medicine, so I think it is worth publishing.

Author response and action taken:

We appreciate and thank the reviewer for the positive comments regarding our work.

-Reviewer Comment 2: Fig2. There should be "regulates" instead of "reguates".

Author response and action taken:

We thank the reviewer for the precise review. We have corrected the word and the edited figure is inserted.  

-Reviewer Comment 3: Fig3. Figures are too small, legend is not visible.

Author response and action taken:

We apology for this inconvenience. As recommended, Figure 3 is now divided into the following two Figures: “Figure 3 and Figure 4”.

Additionally, the legend of the original Figure is now separated and edited, and the new Figure citations are edited in the revised manuscript.

Round 2

Reviewer 1 Report

The Authors did a great job in revising their manuscript.

One concern remains on the ROC Curves: please calculate the Youden’s index (quote PMID: 32964011, PMID:31771147) and report the highest J value in Figure 4.

Author Response

Subject: 2nd round of Revision on a manuscript entitled “Nourin-Dependent miR-137 and miR-106b:  Novel Early Inflammatory Diagnostic Biomarkers for Unstable Angina Patients”

Our response is detailed point-by-point below. Revisions made to the manuscript text, in response to the comments, have been added in page 13 in the marked manuscript using track changes.

 Reviewer 1

Response to Reviewer Comments

 I- Reviewer 1.

 -Reviewer comment 1: The Authors did a great job in revising their manuscript.

One concern remains on the ROC Curves: please calculate the Youden’s index (quote PMID: 32964011, PMID: 31771147) and report the highest J value in Figure 4.

Author response and action taken:

We thank the reviewer for the kind comment about our revised manuscript and for this observation which will prove the validation of miR-137 and miR-106b-5p as diagnostic markers in ACS patients. We calculated the Youden’s index and the J value for miR-137 and miR-106b-5p in discriminating UA patients from heathy individuals (Fig. 4a), as well as between UA and STEMI patients (Fig. 4b). The calculated Youden’s Index and J value are presented below and are added in the text in the results section (page 13), the edited text is highlighted in the text by track changes.

  • miR-137 in discriminating UA patients from healthy subjects: The Youden’s index J=0.91, J value=195.4
  • miR-137 in discriminating UA from STEMI patients: The Youden’s index J=0.58, J value=157
  • miR-106b-5p in discriminating UA patients from healthy subjects: The Youden’s index J=0.75, J value=157
  • miR-106b-5p in discriminating UA from STEMI: The Youden’s index J=0.76, J value=175.

Round 3

Reviewer 1 Report

~